# The Image of Violence and the Study of Material Religion, an Introduction

## Lucien van Liere

Research Institute for Philosophy and Religious Studies, Utrecht University, Janskerkhof 13, 3512 BL Utrecht, The Netherlands; l.m.vanliere@uu.nl

**Abstract:** This article studies the intersection of religion, materiality and violence. I will argue that pictures of violated bodies can contribute substantially to imageries of religious bonding. By directing attention towards the relation between pictures of violence, religious imagery and materiality, this article contributes to current research on religion-related violence and on material religion, two disciplinary fields that have not yet been clearly related. By focusing on the picturing of (violated) bodies as both sacred and communal objects, I will make clear how pictures of violence relate to social imageries of (religious) communities. Two short case-studies show how pictures of violence are recreated in the imagery of communities, causing new episodes of violence against anonymous representatives of the perpetrators. This article develops a perspective on the role pictures play in framing religious conflicts, which is often neglected in studies of religion-related violence. The study of religious matter, on the other hand, could explore more deeply the possibilities of studying the medialization of contentious pictures of human bodies in the understanding of conflicts as 'religious'.

**Keywords:** religious violence; material religion; bodies; pictures; images; iconoclasm

## 1. Introduction

The study of religion-related violence has been strongly focused on religious ideas, theologies, macro-narratives and socio-political contexts. These approaches, though helpful in understanding how and when violence is used by religious actors, often start from a point of view that includes power-analysis, causality and goal-analysis. This runs parallel with popular perspectives within the field of conflict studies. What many of these wide approaches are often missing, however, is a thorough analyses of the materiality and narrativity that can frame and create the dynamics of violence. The narrative dimension of religion-related violent conflict has recently gained some attention from some scholars studying the narratives of terrorists and actors of religious-political violence (Kippenberg 2011; Halverson et al. 2011; Glazzard 2017). The material side of religion-related violence, however, has gained little attention, and is often limited to—although very interesting—case-studies on the materialities of martyrdom (for example: Cook 2017; Pannewick 2020) and on iconoclasms (for example: Morgan 2005; Latour 2010; Mitchell 2011; Noyes 2013; Kayman 2018).

In this introductory article, I will explore how materiality functions in religion-related violent conflict. The leading question is how the analysis of religion-related violent conflict can benefit from a material approach, and how the research field of material religion can benefit from the analysis of religion-related violent conflict. Because materiality, religion and violence have not been clearly cross-linked, I will discuss some of the intersections where these subjects meet. Following scholars like Mitchell, Alexander and Belting, I distinguish between picture and image. While a picture is mere form, sound, color, shape, etc., an image refers to the socialization of a picture in the wide contexts of the viewers. How images work in socializing pictures depends on a complex process of online and offline representations and medializations. In this article, I will pay special attention to the representations

and medializations of human bodies. More specifically, while pictures of bodies are abundantly present (or remarkably absent) in many religious traditions, I will address the question of how pictures of violated bodies contribute to an imagery of violated communities. Two short cases will be discussed, in which acts of violence copy pictures of violence unto their victims. This means that understanding what online or offline pictures of violence 'do' is not purely an academic question, but is also highly relevant for developing strategies for the prevention of violent conflict.

## 2. Religion-Related Violence

Many studies on religion-related violence use wide terminology to understand the specific roles of religious symbolisms, ideas, convictions and doctrines, or to identify certain master conceptual frames that are shared by radical religious actors who do not shy away from violence in order to reach their goals. Some of these frames are used as general models to predict violence against populations or states, such as 'cosmic war' (Juergensmeyer 2000, 2004), 'monotheism' (Morgan 2005, p. 117; Assmann 2008, pp. 106–27; Beck 2010, p. 44; Chirot 2011, pp. 4–5), or the moral and political duality of good and evil (Juergensmeyer 2000; Selengut 2003). Often, religion-related violence is understood as a divine duty (Hoffman 2006). These understandings run the risk of a reification of religion or, in a more popular frame, to understand 'religion' as the box of Pandora. This, despite the fact that the ambivalence of religion concerning violence and non-violence has been widely acknowledged, especially after Scott Appleby published his important study on the 'ambivalence of the sacred' just before 9/11 (Appleby 2000; Juergensmeyer 2004; Selengut 2003). More recently, some scholars work less with master conceptual frames and more with socio-psychological perspectives, using more lucid and pragmatic descriptions of religion (Kippenberg 2011; Meral 2018).

In terrorism-research (as in conflict studies in general), master conceptual frames are often used to identify how people are mobilized behind a religious or ideological cause. Often, terrorism is understood as target-driven (see, for example: Sedgwick 2004; Hoffman 2006; Gregg 2014), while nuanced interpretations have also seen the light (Bhatia 2005; Gunning 2007; Gunning and Jackson 2011). Some scholars, most prominently Randall Collins (2008, 2015), have criticized the alleged target-oriented feature of violent conflict, and directed attention towards the situational embeddedness of violence. This does not, however, mean that the imagery of communal belonging cannot be an important frame for understanding violence. In an important study, Fiske and Rai (2015) argued that most violence is an attempt to confirm or restore social relationships. This means that most violence contains a virtuous perspective on what is important for communities to function (see also: Berman et al. 2019). The material portrayals and medialized pictures of communities suffering gross violence at the hands of religious or ideological opponents are seldom understood as core elements in the violent mobilization of religious actors, but can, as I will argue, be seen as important elements in construing conflict-positions. In this article I will explore how medialized pictures of violence, seeing violent pictures offline or online, images of suffering, narrating violence through pictures of violated bodies, etc., can become incentives for (justifying) violent actions. Indeed, how subjects relate to these images depends on complex local and globalized representations (see for example: Lazarus 2019). However, the imagery of violence is an important construct in the formation of religious affiliations.

'Religion' as an explanatory model for violent conflict, or as Pandora's box, has been sternly criticized by scholars such as Asad (2003), Cavanaugh (2009), Gutkowski (2014), Francis (2016), and many others. Pointing to a strong western bias that religion is something separate from politics, and is confined to the domains of conviction and spirituality, they criticize the term 'religion' as such, and direct attention towards a more complex analysis. Using a too simple understanding of 'religion' in the study of religion-related violence runs the risk of getting stuck in a monocausal analysis. In particular, Francis' suggestion of using a different term for 'religion' defends a stronger social interpretation in the study of religion-related violence, and suggests we look closely at what is 'dear and non-negotiable' for specific social groups. Francis (2016, p. 913) suggests the term 'sacred' to

be useful, from a Durkheimian point of view (see Durkheim 2001, pp. 140–52, 239–42). Other scholars are also critical of an understanding of religion as a monocausal source of violent conflict, and blur the boundaries between religion, politics, ideology and economics (Cavanaugh 2009; Gutkowski 2014; Buc 2015, pp. 144–51).

Focusing on the aspect of materiality as a primal focal point of conflict-analysis opens up ways to understand matter as part of social conflict-positions. Besides, this focus also creates possibilities of seeing that not only can religious icons and special things be part of conflict-causes and conflict-dynamics, but also that certain material portrayals of the community as harmed or violated can heavily contribute to the incitement of violence. As Alexander (2012, p. 16) points out while writing on the social construction of trauma, 'events are one thing, representations of events quite another'.

## 3. Religious Matters

Since the 1980s and 1990s, a growing awareness of materiality within social studies has criticized the idea (popularly ascribed to Durkheim) that social relations can only be explained by social structures, and that materiality is at best a side effect of social relations (Murphy 1995; Schatzki 2010). Nowadays, however, arguing that materiality is at the heart of many religious traditions is so self-evident that this can hardly be seen as a statement (see: Coleman 2009, p. 360). In an interview with Birgit Meyer, David Morgan describes 'religions' as

> fundamentally embodied, material forms of practice in which the coordinates of social life such as gender, power, class, value, and social relations are defined and experienced in material terms. The shape of materiality is webs. Objects, spaces, and people are nodes within these webs that mediate the relations between individuals, groups, and entire networks. (Meyer 2020)

The so-called 'material turn' that has affected religious studies since the 1990s has its roots partly in Foucault's analysis of political embodiment (Foucault 1975), but also in the emphasis on object agency, and in a growing awareness of the entanglement of the natural world in social practices (Mukerji 2015). This 'turn' asks for a deeper study of the 'life of things', the materiality of social religious life, iconic power, and material regimes of power (Latour 2007). In the field of media studies and visual studies, the relationship between the image and the social world of the viewer has been addressed many times, often studying the impact of the image on the viewer (Elkins 1996; Mitchell 2005; Young 2010; Belting and Dunlap 2011). Since the turn of the last century, many of these insights from scholars working on materiality have been made fruitful for the study of religion. This has resulted in more emphasis on the abundant physicality, materiality, and thingness as media of religious communication, conviction, faith and belief. Scholars such as Morgan (2005), Chidester (2000, 2018) and Meyer (2010, 2012, 2019) have conducted constructive and interesting research into the different modes of what is now called 'material religion'. On the other hand, however, only a few authors who have taken seriously this 'material turn' within religious studies have also demanded attention for the relationship between materiality, religion and *violence* (Morgan 2005; Lazarus 2019). The edited volume *Key Terms in Material Religion*, for example, published in 2015, does not pay attention to this subject (Plate 2015). This means that the question raised by Frances King in 2010 still stands:

> The ubiquity of images in contemporary society has not been matched by increased understanding of how they "work", how they can influence behavior, and why people respond to them in the ways that they do. We need to ask what it is about images that both attract and repel. Why some people reach out to touch a statue that others might seek to deface. How some pictures stimulate mob violence on the one hand and acts of veneration on the other'. (King 2010, p. xiv)

This, while many studies on material religion point to matter that is disputed or to things that offend (for example: Meyer et al. 2018). Often, works of art are discussed that transgress the boundaries of interpretation, but also pictures are studied that spark controversies and raise tensions in societies, or are taken out of their places deliberately to provoke groups or communities (Baumgartner 2018; Tamimi Arab 2019). We can think of academic reflections on the controversies raised by Andreas Serrano's photograph 'Piss Christ', published in 1987 (Shine 2015), or the Danish cartoons that were published by the Jyllands Posten in September 2005 (Veninga 2014).

The question raised by King above, of why people respond to pictures the way they do, depends, as shown by Lazarus (2019), at least partly on deep understandings of local powers and histories. Based on a thorough case-study of cybercrime in Nigeria, he found that 'contemporary manifestations of spirituality in cyberspace (life-online)' reflect 'local epistemologies and worldviews in society (life-offline)'. In a broader context, the digital dissemination of information and of visual material can transform (local) perceptions, or create new online communities and loyalties outside the influence of official authorities (see also: Sbardelotto 2019). Some have argued that medialization remodels or transforms, to a certain extent, existing religious patterns (Bratosin 2016). These developments and reflections are important for understanding the impact of medialized pictures on conflict-situations. An important dimension of current-day violent conflict is its cyberlife. Current conflicts do have a significant online presence, which means, as Birgit Bräuchler suggests (Bräuchler 2013, pp. 343–54), that most current conflicts can only be understood by integrating online and offline research. In a similar vein, Meral (2018, p. 21) points to the dynamics in which local incidents are picked up by transnational networks with clear religious or political agendas, and who subsequently disseminate their messages globally. Often, these 'local incidents' that Meral writes about become part of a historic continuum of suffering communities, be it Christian, Islamic, or other. Indeed, the internet plays a growing, significant role in these conflict-dynamics, as specific materialities (pictures, artworks, short videos, memes, etc.) are used to communicate antagonistic messages. The facility of the internet also creates new forms of provocative contestations. The threat of pastor Terry Jones on YouTube to burn 2998 Korans to 'honor the victims of 9/11' in 2013, the online beheading of 21 migrant workers by Islamic State in January 2015 dressed in Guantanamo Bay garb, or the recruitment video 'Oh oh Aleppo' uploaded by Dutch 'fighting journalists' in the same year showing the destroyed homes of Syrians in Aleppo, are just a few cases that show how mediated materiality marks, and signifies, messages through victim-representations to generate shock and attention.

Since the (CNN) coverage of the First Gulf War in 1991, academic attention on the medialization of war and violent conflict has been growing. First, this attention consisted predominantly of (philosophical) reflections on medialization and representation, and on 'the real' versus 'the virtual', but later a critical appreciation developed for how medialization is part of (violent) conflict itself (Bunt 2003; Karim 2003; Lewis 2005; Zarkov 2007; Marsden and Savigny 2009; Tulloch and Blood 2012; Bräuchler 2013), and for how the internet functions in creating new socio-religious agencies outside the scope of institutionalized religious authorities (Hjarvard 2008; Sbardelotto 2019). Often, gripping and rapid-spreading media-images of conflict contain injured or dying human bodies that affirm or change epistemologies of war and conflict. In state-based counter-terrorism initiatives on Facebook and Twitter, counter-narratives are often accompanied by pictures or short movies, with the purpose of delegitimizing religious radicalization. The medialization of suffering is indeed part of politicized, ideological or religious argumentation. But this process contributes highly to the dramatization of conflict through (re-)creating the iconic (Tulloch and Blood 2012), and is often charged with historic stereotypes (Van Liere 2020). Indeed, materiality plays a decisive role in medialized conflict-positions. However, the relationship between pictures and social networks, regarding how they 'work' in instigating social violence, how they can encourage acts of violence and stimulate efforts to 'copy' the disturbing picture into real, offline situations, has not yet received the attention it deserves.

## 4. Special Things, Places, Communities

Within religious traditions, the 'knowledge' that pictures do something seems to be common sense; from the relics in Roman Catholic Christianity to the tangible materiality of martyrdom in Shia traditions, and from the statute of Buddha in Theravada Buddhism to the empty walls and open Bible in many Protestant churches. Materiality refers, unites, and shapes religious imageries. Color, shape, sound and form are—so to say—part of the content of religious traditions. Materiality is part of the communal mediations of religious groups, be it as moveable stuff, or as a fixed place with a name. In the study of religion, places, as both geographical locations and as ideas, have rightfully been given much attention. Religious places are often sites of memorialization, where the past becomes reactivated in narratives and emotional testimonies, while some are endlessly copied into wearable, movable objects. The emotional energy that is related to places like Karbala in Shia, or Lourdes and Guadalupe in Roman Catholic traditions, brings millions of people together annually. While they function as special sites, some are also contested objects of competing narratives, like Medjugorje (Herrero Brasas 2007; Wiinikka-Lydon 2010), Ayodhya (Van der Veer 1987; Friedland and Hecht 1998) or the Temple Mount (Friedland and Hecht 1996). These contestations are not just efforts to sacralize or desacralize certain places, but are also strongly amalgamated with memories of, and narratives about, socio-religious conflicts, social constructions of victimhood, and claims on the right to be.

What things and places mediate or (re)present depends on the social activities they are embedded in and part of. They are—so to say—complex accumulations of social histories, narratives, memories and convictions. Special things can be understood, in a Durkheimian fashion (Durkheim 2001), as charged objects of social-religious engagement that mirror social energy as these objects circulate in narratives, rituals and memories. This means that things can 'make themselves feel' through their gaze, shape, color and smell. Within religious traditions, special things are often understood as iconic, as deeply rooted objects of social experience that contain a certain subjectivity. When writing about 'the iconic', Jeffrey Alexander contends that the 'iconic is about experience, not communication. To be iconically conscious is to understand without knowing ( ... ). It is to understand by feeling, by contact, by 'the evidence of the senses', rather than the mind' (Alexander 2008, p. 782). Objects become icons when they do not only contain material force, but also have symbolic powers. 'Actors', Bartmanski and Alexander argue, 'have iconic consciousness when they experience material objects, not only understanding them cognitively or evaluating them morally but also feeling their sensual, aesthetic force' (Bartmanski and Alexander 2012, p. 1, see also Qin and Song 2020 on the power of Buddhist symbols).

Sacred things are not fixed within religious traditions, not even if they are 'iconic'. Birgit Meyer et al. write that objects circulate, and that scholars should be attentive towards their local adaptations, to what people do with them (and to them), and to 'the affective and conceptual schemes whereby users apprehend an object' (Meyer et al. 2010, p. 209). As things circulate, meanings circulate, concepts change, and feelings move. Things do have 'lives', as W.J.T. Mitchell playfully argued in 1996, teasing the Kantian *Wende* (Mitchell 1996). They have, in the words of Meyer et al., 'social careers' (Meyer et al. 2010, p. 209). The 'curriculum vitae' of a thing contains references to the histories in which it has been charged with sensual, aesthetic force in different ways. Religious things are, so to say, charged and recharged with the social energy of a community, which can link, through sensual objects, to a (imagined) past, and to a beyond. The community senses the relationship of things to themselves, while objects mirror social relations. They function in narratives while narratives are told through things and places. As such, they contribute to the concrete experience of a community. David Morgan, echoing Benedict Anderson's famous thesis on imagined communities (Anderson 1991), argues that members of communities need 'symbolic forms such as songs, dance, images, and food to allow them to participate in something that is larger both spatially and temporally than their immediate environment' (Morgan 2005, p. 59). This concrete embodiment of community experience marks the sensation of the larger community itself as the subject of religious symbols and rituals. In this way, things contribute to the imagery of what is 'dear and non-negotiable' to a

community. They contain—in a certain respect—the 'nerves of the community', which gives them meaning. Because they mirror, affect and mediate the community, things play important roles during violent, religion-related communal conflicts. It is vital for my argument not to confine 'religious things' to sacred objects with iconic reputations within religious traditions, but to extend the interpretation to things and pictures that evoke the—often physical—image of the religious community.

## 5. How Things (Might) Work in Socio-Religious Contexts

As mentioned above, the difference between image and picture that has been pointed out by several scholars is important for our subject. In an interview in *Image and Narrative*, W.J.T. Mitchell describes a picture as

> at least potentially a kind of vortex, or "black hole" that can "suck in" the consciousness of a beholder, and at the same time (and for the same reason) "spew out" an infinite series of reflections'. (Grønstad and Vågnes 2006, p. 1)

One year earlier, Mitchell published *What Do Pictures Want* in which he describes pictures as 'complex assemblages of virtual, material, and symbolic elements' (Mitchell 2005, p. xiii). As such, pictures function as powerful tools at all levels of cultural and religious representation. However, a purely material perspective would have to deal with the question of why some materiality is so 'powerful', why some things evoke so much outrage, like headphones put on a Buddha statue in Myanmar (Clark 2015). Pictures, Mitchell writes, are not only individual things, but they have also a symbolic form 'that embraces a totality' (Mitchell 2005, p. xvii). This is what he calls an 'image'. An image is 'any likeness, figure, motif, or form that appears in some medium or other' (Mitchell 2005, pp. xiii–xiv). The 'apparition' of an image in a picture is an important key to understanding why a focus on materiality is significant for the study of religion-related violence. The image can be understood as what 'happens' to humans at the surface of a picture; a complex flow of cultural memory, collective history, local epistemologies, religious representations, cultural narratives, (past) grievances, remembered and narrated histories of suffering or victory, but also of individual experience, etc. At the surface of the picture, this all debouches into physical feelings of joy or anger, uncanniness or frightening awe, comfort, pleasure or shock. In a similar vein, Jojada Verrips directs our attention towards physical responses to pictures of sex and death, as he argues that humans are involved in 'a continuous process of storing, retrieving, and re-combining sensations, emotions, and knowledge in the body' (Verrips 2018, p. 302).

How people respond to a picture depends on the appearing image that comes alive through the picture. Destroying an image is impossible. Only pictures can be destroyed. In the aforementioned interview, Mitchell argues that

> the image survives ( . . . ) destruction, and often becomes even more powerful in its tendency to return in other media, including memory, narrative, and fantasy ( . . . ). The act of destroying or disfiguring an image ( . . . ) has the paradoxical effect of enhancing the life of that image. An image is never quite so lively as in the moment when someone tries to kill it. (Grønstad and Vågnes 2006, p. 3)

Mitchell's understanding of picture and image can only be understood if we apprehend how compound and unstable the image is, and how the destruction of a picture contributes to the intensity of the image that is 'felt' by a complex meaning-giving community.

Belting (2012, p. 187) writes, in line with this, that images emerge in the act of looking (see also: Hillis Miller 2008, p. s59). Looking is, he continues, as a facility of the body, the carrier of our entire knowledge about images. If understood correctly, we are not looking at an image, as if there are two separate worlds (one of looking and one of the image), but images form in adapting our looking, which Belting calls 'the gaze'. 'The complicity between body and gaze leads to the image' he writes (Belting 2012, p. 187). The gaze is, so to say, also displayed in the image, so it seems that images have

the ability of looking back, or of answering our gaze. Belting's understanding of image is different compared to Mitchell's, but his notion is especially interesting for pictures of violence and suffering, as these are used throughout modern media. I will expand the notion of the image to a more general socio-religious frame. Images have, so to say, social lives, and the gaze is embedded in social networks of memory and solidarity. The curriculum vitae of a thing, as I mentioned to above, is filled with vague and sharp references to collective and individual episodes, narratives, rituals and memories that appear in the image and relate to social bonding. The image can also burst out of the thing's frame. This is what can happen in medialized suffering and violence. Echoing, for example, the iconic power of the image of the gazing crucified Christ drawing a 'pro nobis' (for us) from the faithful Catholic, many pictures and pamphlets in western media can be understood as secular efforts to arouse a similar hamartiology among those who are 'forced by the picture' to look. For example, the Abu Ghraib picture of a hooded human figure wearing a black cape and standing with his arms spread with wires attached to his hands combines a complex image of a Christ-like pose, a Klu Kux Klan-like dress, and a fascination with victimhood (Mitchell 2011). Pictures of the murdered Matthew Shepard in 1998 raised, in the complex cultural situation of 'culture wars' in the US, the imagery of the crucified Christ (Middleton 2020, pp. 190–95). Both pictures became iconic assemblages. It is highly doubtful that these pictures would have been so strongly medialized and re-created in all kind of artistic interpretations without this iconic power.

Pictures can play a significant role as specific conflict-positions. The gaze can be evoked, and although it is always uncertain what images are brought to life, the impacts of pictures of suffering, death or violence are often severe, and relate the bodies of the gazing humans to these pictures through shock, shivering or exalting sensations (Van Liere 2015). Although the gaze can never be totally controlled, picturing violated human bodies, especially those of children, can constitute an effort to frame the effect of the gaze, as these pictures evoke an imagery of the responsibility to act and protect (Tulloch and Blood 2012, pp. 31, 60, 77, 83). This is certainly not new, although the picturing of victimized and martyred bodies does have a history. To give just a few well-known examples from European history: During World War I, young men in Britain were 'addressed' by strongly gendered posters with women and children, encouraging them to go to the front (Goldstein 2001, p. 272; Mitchell 2012, p. 36). A few years later, during the Russian famine (1921–2), western European audiences were confronted during campaigns with pictures of children who were close to starvation, pictures that were, as Fuyuki Kurasawa noticed, deliberately used to stalk the conscience 'of subjects who, through the very process of seeing images of suffering, are constituted as moral audiences upon which is thrust the burden of responsibility to alleviate it' (Kurasawa 2012, p. 69). Kurasawa's note applies to many contexts, as examples can easily be found in current times. Although the effect of pictures of (extreme) suffering has not changed, the dissemination of pictures through modern media to frame and change a conflict has developed very quickly, as governments and conflict actors are moving online and expanding conflicts into cyberspace (Keenan 2012; Bräuchler 2013; Meral 2018). The shivering body is often evoked in contexts of conflict propaganda, by war and conflict information media, and during humanitarian interventions (Zarkov 2007; Keenan 2012). Pictures of suffering strongly touch the nerves of a community and arouse the image of what is 'dear and non-negotiable'. Many actors of violent conflict or fierce protest are performative witnesses, having been touched by the gaze. They refer to medialized pictures and narratives of suffering while justifying their violent actions. From the texts of Osama bin Laden, referring to 'women and children' in Lebanon (Glazzard 2017), to Brenton Tarrant (responsible for the attack in Christchurch on 15 March 2019), referring in his manifesto to the Stockholm attack in 2017 that killed a 'young and innocent' 11-year old girl. Pictures of broken bodies gaze at the subject and communicate through the physical attunement of shock and unsettlement. The enduring strength of the touch depends on the power of the imagery of suffering that 'reveals' what lies beyond the picture. The body acts as both medium and image (see: Belting 2012, p. 191). This brings us back to the question of religious matter and violent conflict. The shivering caused by cartoons of the prophet Muhammad, the shock and anger raised by the

burning of the Quran on a barbecue, the sadness felt seeing the spitting on the crucifix, the insult felt seeing a Buddha statue with a headset, etc., are part of the wide complex of medialized communal solidarity, memory, rituality, bonding and individual experience. This is why these cross-border actions cause physical responses. They touch the nerves of a community by emptying the energetic link between its members that has charged the icon, and which becomes intensively felt at the moment of its humiliation or destruction. It is surely no coincidence that many religious icons are pictured as human and humanoid forms, as they attune the looking body to the body of belief and community. Rejecting a dualistic interpretation of body and picture as subject and object, Wendy Lee contends that 'bodies are linguistic: posited as canvas for cultural and political inscription, bodies are neither merely canvas nor mirrors, but rather sites of inscription, exchange and regulation, dissent and satire' (Lee 2005, p. 289, see also: Belting 2012; Verrips 2018, p. 302). The picture is 'dressed' in the physical image of the community.

## 6. The Human Body and the Image of the Sacred

In his playful book on visual arts, Elkins (1996) writes that the world is 'full of eyes', and that 'sight is everywhere' (p. 75). He describes 'extreme experiences of seeing', which can shock and make us look away: the sun, genitals and death (p. 87). Elkins contends that we organize our seeing around the seeing of bodies. There is a kind of desperation to our desire 'to see faces and to read them as expressions of mental states' (p. 192). We seek bodies in the unfamiliar object (p. 129). What Elkins describes can be widely applied.

In many religious traditions, materiality relates heavily to the human body. Not only in anthropomorphic figures, pictures, prayer cards of saints, Jesus on the cross, pictures of martyrs, relics, statues of Buddha, the imaginations of physical pain in hell, human-like frightening beings like demons or in the humanoid beauty of angels or apparitions of Mary, ghosts and deities, but also in certain physical ritual gestures, special dresses and food, like the Roman Catholic host, that may or may not be eaten or that is used during certain rituals, or the prohibition of picturing the body of the prophet Muhammad. The material(ized) body-picture is part of a process of fusing the gaze with images of nearness, embeddedness and meaning, but also with social bonding, network and memory. The body pictures, so to say, the image of the community. On the other hand, however, these body-pictures have been broken, smashed, taken out of their places and demolished as idols, or as (re)presentations of hostile communities or faiths, not only as part of theological critique, but also as consequences of social conflict.

Let us look at a historic case that can elucidate these dimensions of physicality and imagery and show the different dimensions in which the body functions. The purpose of this short detour is to show how 'things' are strongly related to the human body, and simultaneously to communal imageries. During a Roman Catholic mass in the royal chapel of the Portuguese King João III, in 1552, an English Calvinist merchant

> suddenly leapt from the crowd, punched the elderly priest and seized the consecrated host from his hands. Before he could be stopped, William Gardiner tore the host in front of the horrified congregation, hurled it to the ground and stamped on it. (Soyer 2019, p. 122)

Historian François Soyer writes that only the personal intervention of the King could prevent the crowd from lynching Gardiner in the chapel itself. Gardiner was later executed in a gruesome manner after a short trial (Soyer 2019, pp. 122–23). João III was well-known in his own community as a very pious man, related to the Inquisition Court in Portugal. Gardiner, after his execution, gained a place in *Fox's Book of Martyrs,* which was first published little more than 10 years after the incident and contained about 1600 pages of Protestant martyrologies, complete with 60 woodcuts, many portraying the violated human bodies of the martyrs.

This incident in 1552 shows different modes of the body: the host as the body of Christ (stories about bleeding hosts are common in Catholic history, Bynum 2004); Gardiner, who took the host out of its

ritual place (the priest's hand), and conducted an 'alien ritual' that charged the object with a different meaning, namely, that of the meaninglessness of the thing and of the idolatry of the community charging the host with the meaning of Christ's body; and—finally—the denotation of Gardiner's destroyed body as a martyr's body, added to the canon of visualized and narrated violence of 16th century persecutions (*Fox's Book of Martyrs* remains popular in orthodox Protestant circles even today).

The host and Gardiner are configured in a serious play, as two separate bodies that become fully charged with the intensified meanings of communities and their interrelated histories. The host is sacred because people communicate through memory, narrative and ritual, and thus participate in its sacred-making (see: Anttonen 2000, pp. 280–81). The host can be understood, so to say, as the charged battery of a community (see: Collins 2004, pp. 81–95), and is loaded with social energy (rituals, narratives) and contextual significance (contestations, affirmations). The sacredness of the host is the community itself that charges the object with its image of solidarity and eucharistic significance. Sacrilege is, so to say, the image relating not only the host to the body of Christ, and not only the action of Gardiner to an act of violence towards Christ's body, but also relating the belonging of the Catholic community to a tense threat of 'outside' Protestants and 'inside' heretics (which was seen as a big issue in Andalusia in 1552). Gardiner's action intensifies the image of the host at the moment of its demolition, discharging the accumulated social tensions upon his body. Subsequently, Gardiner's body can also be understood as a charged object that appears in the image. His body becomes the object through which justice becomes manifest and meaningful. His execution is not simply the liquidation of an annoying culprit, but a manifestation of the law that binds the community. Philosopher of Law Paul W. Kahn speaks about the 'transubstantiation' of the sacred in the 'flesh' of the individual, whereby a body as a material 'thing' loses its finite character and becomes the place where 'the sacred' becomes manifest. The sacred, in his understanding, is carved as meaning in the flesh of the individual. Thus, the finite human body is meaningless if it does not function as an instrument through which the sacred becomes manifest. According to Kahn, the 'sacrifice' is the central sign of loyalty to the sacred (Kahn 2008, p. 32). This means that the imagery of social connectedness becomes manifest in the act of executionary violence. The violence done to the body can only proceed if the body of the community as medium creates the image of the violated host. In a similar vein, Gardiner's transformation into Protestant martyrdom, and his medialized presence in northern Protestant communities, also signifies the carving of the meaning of the sacred into the flesh of the individual. His narrative has become part of a canon of transgenerational medialized suffering, that copies the meaning of his action and death each time the story is read.

The relation between icon and body is important for our understanding of religion, materiality and violence. Gardiner's suffering implodes in the iconic suffering Christ (a trajectory of the martyr). His body belongs to the community as Christ's body, and becomes iconic when it is destroyed. Picturing suffering, humiliated, broken, dead bodies evokes physical responses, from looking away to shivering and shock, and brings the onlooker into the vicinity of the sacred. Suffering can re-bond the community, while activating the sacred as what is 'dear and non-negotiable'.

## 7. Crying Pictures and Charging Bodies

Until now, we have pointed out how the image is a complex bricolage, and how pictures of violence evoke the subjective gaze within the image. In the following two cases, to which I will give much less attention than they deserve, violence, materiality, sacredness and religion collapse in the image. The first case offers an interesting witness to the process of how pictures come alive, and how the subject is touched by the object's gaze. The second case explores how the gaze is evoked through style and image, and opens memories of violence in the act of violence. Both cases show how images circulate in a global forum through medialized pictures, how complex histories, power and religion (as part of specific communities of belonging) are displayed, how conflicts have become cyberwars with offline consequences, and—finally—how bodies become dense quotes of complex histories of violence.

The first case comes from Indonesia. Imam Samudra (Abdul Aziz) was one of the Bali bombers responsible for a double attack on two bars in Kuta, Indonesia, in October 2002. The attack took the lives of at least 202 people (see: McIntyre 2016). In his notebook, Samudra narrates how he was surfing the internet and was hit by pictures of the bodies of dead and injured children in Afghanistan after the US bombing of Kabul in November 2001. The human costs of the first American strikes on Kabul were widely published by news agencies and on the internet. Samudra describes how he became unsettled by photos of dead children. In his notebook he wrote:

> Those images are photos of what really happened, that are scanned, put into a computer, and then uploaded onto the internet. They are immovable, without sound, numb. But the souls cried out in agony and their suffering filled my heart, taking on the suffering of their parents . . . (Tempo Editors 2003, p. 15)

In a subsequent diary-phrase, Samudra wrote: 'Your weeping, oh headless infants, slammed against the walls of Palestine, Your cries, oh Afghani infants, all called to me; all you, who, now armless, executed by the vile bombs of hell' (Tempo Editors 2003, pp. 15–16). It is clear that these medialized pictures are more than mere figurations. Samudra acknowledges the mere materiality of what he saw ('immovable, without sound, numb'). Yet, by using the conjunction 'but', he describes how pictures that are 'immovable, without sound, numb' come alive, how he heard 'their souls' cry out in agony. This conjunction figures the moment of shock Elkins describes, the imprisonment of Samudra's looking by the dead children. The experience of war pictures has already been described by John Berger (reflecting on Donald McCullin's war photography) as being 'arresting' while utterly discontinuous with normal life. "We are seized by them", Berger wrote in 1972, while pointing to the 'double violence' of photography—the violence of war shown by the picture, and the violence of the moral inadequacy and moral inability the viewer starts to realize while seeing the agony that is portrayed, dispersing her sense of shock (Berger 2013, pp. 31–33). But not so Imam Samudra. Samudra gains his subjectivity to overcome the arrestment. As a response, he starts to talk to the pictures as if they were persons, and begins to understand the pictures as media for wider circles of suffering in Afghanistan and Palestine. The image of the suffering of Muslim communities in Afghanistan and Palestine becomes active as a stringent, inescapable frame. This image relates religious ideas and (imagined) relationships to the pictorial media-production of massive suffering. The 'gaze' of these communities is strongly felt by Samudra as a sacred demand for action. As the pictures cried out to him, he implodes, so to say, in the intimacy of the children's parents, and explodes in the grand master narrative of Muslim suffering. The image that is alive in these cries contains wide interpretations of global conflict, 9/11, and the attack on Afghanistan in November 2001, but also activates the theological language of judgment: the suffering was caused by the vile bombs of hell.

By bombing two clubs in Kuta, Bali in 2002, Samudra became an actor in the atrocities he observed (Van Liere 2015, p. 17). The suffering of the pictured children configurates his insouciance towards his victims (Samudra 2004, pp. 155–58). The dead bodies of the children become religious war-territory. The image that came alive in the act of looking is directly connected with imbalances in power, injustice, cruel over-powerful western forces and victimized Muslims, a thematic network he explains in his book *Aku Melawan Teroris* (Samudra, see also: Bin Hassan 2007). What Samudra's case illustrates well is how medialized pictures of suffering circulate across the globe, and how these pictures are decontextualized from their local atrocities and recontextualized in an attack that projects the images of a violated fragile community upon the symbolic representatives of the perpetrators by taking violent revenge (see: Van Liere 2017).

Another case that illustrates what pictures of violence do, and how they can be part of communal conflict positions, appeared in the slipstream of the 'Abu Ghraib' case. The pictures that were widely published in April 2004 of the abuses in the Abu Ghraib prison in Bagdad have already been widely discussed, including from material religion and iconographic points of view (Lincoln 2007; Mitchell 2011, pp. 137–59; Paul 2011; Keenan 2012). As is well known, the photos showed prisoners in (sexually) humiliating positions, sometimes in Abu Ghraib's orange prison attire, sometimes dressed in

black cloths, but often naked. Lewis (2005, p. 237) points to the fact that the rapid online dissemination of the pictures was because the pictures combined the vital ingredients of sex and violence within a moral and political context. In May 2004, a few weeks after the abuses in Abu Ghraib prison in Baghdad were extensively reported by media, a video was uploaded onto the internet showing how a young man dressed in orange clothes was decapitated. The video, published on the website of *Muntada al-Ansar*, was titled 'Sheik Abu Musab Zarqawi slaughters an American infidel with his hands and promises Bush more'. Al-Zarqawi was the Jordanian leader of Al Qaeda in Iraq.

Applying insights from the microsociology of violence, as developed by Collins (2004, 2008, 2015), to the video, we can see how the actors are performing a ritual that is repeated in many other videos that would follow. To overcome confrontational tensions, differences between perpetrator, victim and bystanders are created. The young man in the video kneels, is dressed in the color of Abu Ghraib's prison clothes, is handcuffed. He is half surrounded by five darkly dressed hooded men wearing guns over their right shoulder. The tension is intensified, and the emphasis is on a clear asymmetry between perpetrator(s) and victim. The sharply designed ritual in the video creates differences by showing black against orange, standing against kneeling, anonymity against face, creating power against fragility. The title of the video is also part of the picture, and creates a difference between biography (Abu Musab al Zarqawi) and category ('an American infidel'). In his thorough micro-sociological analysis of violent action published first in 2008, Collins shows how the asymmetry between victim and perpetrator is often created in social conflict in order to overcome confrontational tension. The well-positioned actors in the video are important. Only one of them commits the act of violence, but the others have, as bystanders, a role that is vital as well. The perpetrator relates to his victim as well as to his peer group. His act of violence is not so much evoked by the victim who is forced into a fragile position, but more so by the bystanders, who create a network of solidarity for the acting perpetrator. Solidarity with the bystanders stuns individual responsibilities (Weenink 2013), which makes the act of violence a collective act. Perpetrators and victims are part of a ritualized encounter, in which the roles of victim and perpetrator are distributed along emotional patterns. The creation of difference in the design of violence (the victim and perpetrator do not face each other) not only overcomes the confrontational tension that Collins analyzes (Collins 2008, pp. 83–99), but is also a representation of the imbalance of power in Abu Ghraib. The victim dressed in the prison clothes of Abu Ghraib evokes the image of humiliation and violence. The online beheading was part of a greater cyberwar in which conflict-actors abundantly disseminated pictures of suffering and violence.

The 'American' in the video was Nicholas Berg, a technician working in the communication industry in Iraq. The website of Muntada al Ansar makes it clear that this beheading is all about Abu Ghraib. US Major Alexander Maxwell (pseudonym) writes (quoted by R. Gordon) that the abuses in Guantánamo Bay and Abu Ghraib have contributed to the growth of Al Qaida more than any Islamic ideology or theology (Gordon 2014, p. 164). Interestingly, in the video, the masked man who takes the lead explicitly refers to 'the shameful photos' of Abu Ghraib as an 'evil humiliation for Muslim men and women in the Abu Ghraib prison'. He continues: 'Where is the sense of honor, where is the rage? Where is the anger for God's religion? Where is the sense of veneration for Muslims, and where is the sense of vengeance for the honor of Muslim men and women in the Crusaders prisons?' (Filkins 2004). Then, the hooded man directly addresses then-US President Bush:

> Regarding you, Bush, Dog of the West, we are giving you good news which will displease you. Your worst days are coming, with the help of God. You and your soldiers will regret the day when your feet touched the land of Iraq and showered your bravery on shelters of Muslims. (Filkins 2004)

On the website, where the speech was published, a well-known Abu Ghraib picture was published that had appeared regularly in Western media in the weeks before, showing a dog threatening a terrified prisoner dressed in Abu Ghraib gear. The US President is visually 'present' in Abu Ghraib's dog. Strongly framed on the website, the picture evokes the imagery of impure, aggressive, western ('crusading') power, and a fragile Muslim community. After the video was published (and quickly

removed), searches for 'Nick Berg beheaded' and variations were for days most wanted on Google (above Britney Spears). In many different contexts, the pictures of 'Abu Ghraib' caused artistic, political and academic responses. In 2014, the movie *Boys from Abu Ghraib*, written and directed by Luke Morgan, was released in the US. The movie was a clear effort to benefit an American public and to come to terms with Abu Ghraib by making the victims less innocent and the perpetrators more understandable; the victims and perpetrators of Abu Ghraib became charged with an American nationalistic imagery of justice.

In an article on iconic violence, Martin Kayman writes about the 'superior capacity of images, relative to verbal discourse, to *present*, rather than represent, the truth—in short, their ability to command an ethical conviction of belief' ([Kayman 2018](), p. 144). This ethical conviction of belief, in the case of Nicholas Berg, is strongly inaugurated with semiotic differences and a strong 'iconic consciousness' evoked by the color and dress of Abu Ghraib. In both cases, the viewer is forced into a mode of extreme looking and looking away. The image becomes fully alive through these pictures of death and is embedded in the sensational experiences of different communities. Nicholas Berg becomes 'America'; the dead children becoming victims of the 'vile bombs of hell'. The destruction of bodies is the obliteration of communities, as the physical picture is part of the imagery of communities. What Samudra heard was not just the screams of dead children and their parents, but—if we can say so—the sound of iconoclasm. What he imagined to be 'dear and non-negotiable' was painfully touched. In the case of Berg, the picture is dressed in the image, while global violent conflict is played out through the bodies at the micro-level of social solidarity with the victim and alienation from the cloaked religious perpetrators in black. But what is most striking in both cases is the way pictures of violence are circulating through the internet, causing real actions of violence against people who symbolically implode into representations of perpetrators. How, in other words, violence becomes iconic through the dissemination of its picturing, and the imagery of victimization and perpetration.

## 8. Concluding Remarks

The cases discussed above show how pictures become charged with images, how they can become dense with history and contribute to the construction of the icon. The attack Samudra co-organized on two clubs in Bali to avenge the medialized pictures of dead children, and the raging 'zeal' in the Muntada al-Ansar video against the 'humiliating images' of Abu Ghraib, contributes heavily to the iconography of 'America'. In both acts of violence, human bodies become material 'things', as they lose their finite significance and become the place where 'the sacred' is carved as infinite meaning in the flesh of the individual (see: Kahn, above). In this sense, both acts of violence refer to the sacred by smashing the icon. If we understand 'the sacred', in the words of Matthew Francis, as 'what is dear and non-negotiable', it becomes clear how the sacred in these cases is charged with trauma-images of pictures of crying children who are reaching out to the subject, and of 'humiliating' photos. In these cases, materiality contributes heavily to the content of the sacred, as trauma-pictures are part of the image. What is 'inscribed' unto the flesh of the victims is the sacred as an infinite law of 'justice' that rejects the image of suffering by inflicting suffering upon the human content of the icon. It is the human body and the destruction of human bodies that give the icon its traumatic power. The icon, however, is not the same as the sacred. 'What is dear and non-negotiable' is the community itself. Its non-negotiability is made tangible by smashing the icon.

How can the study of religion-related violence contribute to, and benefit from, a material religion approach? In many studies on iconoclasm, there is an inclination to understand iconoclasms as strategies that are goal-oriented, like purifying or proving the emptiness of the icon, or articulating untruth. This runs parallel with a dominant current in conflict studies and with the analysis of religion-related violent conflict. Conflicts are often understood in line with what Chris Mitchell formulated in 1981 as 'any situation in which two or more 'parties' (however defined or structured) perceive that they possess mutually incompatible goals' ([Mitchell 1981](), p. 17). Although 'goals' may be defined incredibly widely, what is often neglected is the regime of the image that is not so much

goal-oriented but focused on the past. To put it bluntly, if living together is a goal, it might become incompatible if this being together is disturbed by crying pictures, or pictures of humiliation that are connected to specific icons of power. 'Religious violence' is often not future-oriented but past-driven. This means that in the study of violent conflict, we should ask how pictures, images and narratives are charging the *perception* of different goals with a sense of incompatibility.

In this issue, Margaretha van Es demonstrates how an object can potentially become 'a means of provocation in a larger entanglement between people and their perceived intentions', and how imageries of this entanglement create tensions and conflict. Objects without intrinsic 'provoking' meanings, like pig meat and a prayer rug, become material tools in what seems to be a theatrical street-play, and function as situationally charged tokens for the playful but serious occupation of territory and the right to be. The incompatibility of pig meat and a prayer rug can only be understood if the different imageries of conflicting communities are taken into account (Van Es 2020).

Indeed, iconoclastic gestures, from the blowing up of Buddhist statues by Taliban militia to the destruction of WTC in New York, and from the removal of statues memorializing communism in Eastern Europe to the demolition of the statue of Saddam Hussein by American military, all have in common the idea that these icons were empty, void casings of idolatry or false ideology. But what should also be taken into account is how these icons were charged with images of violated communities; the Russian suppression of Afghan tribes, the American presence in the Middle East and its support for cruel regimes, the communist repressions of dissidents in Eastern Europe, all have created imageries of suffering and oppressed communities. The 'iconoclasts' who flew into the WTC Towers had charged the icon with narratives. Osama bin Laden's letters and interviews are not so much witnesses of fierce iconoclast doctrines (his theology is rather weak and eclectic), but of narratives circling around the construction of victim-identities. His writings are full of images of violence, pointing to an experience comparable with Samudra's (Cullison 2004; Ibrahim 2007, pp. 161–71; Glazzard 2017). Iconoclastic gestures cannot be fully understood if the narratives and pictures of violence, threat and distrust are not taken into account.

At the same time, the human body is used in pictures and icons as the site of political and religious inscriptions. We have to ask how picturing the human body contributes to charging the power of the image, how matter becomes the nerves of the social body, and how the attunement of the violated pictured body and the shocked viewer's body can be understood. It is clear that Imam Samudra 'feels' the pain of the headless children, and can imagine the pain of their parents. Material bodies are strong sensational forms that (re)position the subject and make the community to which the subject relates tangible.

Directing more attention towards material religion in the study of religion-related violence is directing more attention towards how pictures are used and images created, and why icons are smashed. It means that less attention is given to doctrines and ideas as sources of violence, and more to how communities produce and reproduce themselves in narratives and pictures. Pictures can reveal specific conflict-positions, and evoke imageries and interpretations at cognitive, moral and emotional levels. Directing more attention towards religion-related violence in the study of material religion means directing more attention towards the interrelatedness of pictures and images, towards the intense lives of pictures of suffering and death within religious communications, and towards how this all connects to power-frames and interpretations of community bonding. It means that we should become more attentive in the study of religion to how pictures of violence circulate online, and to how humans make twisted copies of pictures of violence in real life situations.

**Funding:** This research received no external funding.

**Acknowledgments:** I would like to thank Margaretha van Es for her constructive feedback on the first draft of this article. Special thanks also for the anonymous reviewers for their helpful comments and suggestions.

**Conflicts of Interest:** The author declares no conflict of interest.

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
