# Peer review of "The Image of Violence and the Study of Material Religion, an Introduction"

_religions, doi:10.3390/rel11070370_

Round 1

Reviewer 1 Report

Thank you for writing this brilliant article! Thus, I would like to commend the authors for the work they have done. However, there are some minor issues that I have with the paper. Consequently, I recommend a revision and I offer my suggestions in no particular order, based on the following points (see the attached PDF document for details):

  • Include pictures, images, figures?

  • Missing citations & references…

  • Abstract

  • Images in contemporary society

  • Material objects & aesthetic forces

  • Paragraphing

  • Proofreading

Author Response

Image of Violence and the Study of Material Religion

Thank you for writing this brilliant article! The manuscript is interesting to read,

original, and publishable, in my opinion. I would, therefore, like to commend the

authors for the work they have done. However, there are some minor issues that I

have with the paper. Consequently, I recommend a revision and I offer my

suggestions in no particular order as follows:

Thank you for reading my article so carefully and responding so enthusiastically to its content! Your comments were very helpful, including the references!

  • Include pictures, images, figures..

The authors wrote:

“This, while many studies on material religion point to matter that is disputed or to

things that offend (for example: Kruse, Meyer, and Korte 2018). Often, works of art

are discussed that transgress borders of interpretation, but also pictures that spark

controversies and raise tensions in societies, or are taken out of its place deliberately

to provoke groups or communities”.

Do you have a few images or figures to strengthen your illustrations and case

studies? Although it is optional to include images or not in this case, but to include a

few figures – pictures, images, illustrations, strategically within the paper, could

help to showcase your contributions in a brighter light, in my opinion.

I have added a few references to studies on the impact of works of arts etc. Although it is a great suggestion to add pictures, I hesitate a bit. Not only because of copy-right issues but also that I think pictures would be most relevant in the final section on the ‘weaping pictures’ of Imam Samudra and on the beheading of Nick Berg. I hope you agree that the references are sufficient here.

  • Missing citations & references…

On page 1, the authors wrote: “Because materiality, religion, and violence have not

often been clearly cross-linked…” Please, if the authors agree, this part of the

sentence requires citations (unless no one has ever said it before this current article).

I skipped the word ‘often’ in this sentence. Although there are works on iconoclasm, these generally include religious intolerance as explanatory frame. What I want to do is to see how the picturing of violence itself relates to religious understandings of conflict. I was a bit in doubt about Morgan (2005) but I am afraid he uses the same pattern. This is why I skipped the word ‘often’.

On page 2, the authors mentioned “Alexander (2012)” in-text (and also in other

places in their manuscript), but it is missing in the reference list. Please, doublecheck

and synchronize the reference list with the in-text citations.

Done. Thank you!

On page 4, the authors wrote: “Special things can be understood as charged objects

of social-religious engagement that mirror social energy as these objects circulate in

2

narratives, rituals, and memories”. I believe a direct acknowledgment of Durkheim’s

(1967) book is needed here, please.

Done. Thank you!

  • Abstract

It is crucial to begin an abstract by briefly stating the purpose, objectives or principal

research questions. You might want to avoid starting your abstract with contextual

statements, such as: “The study of religion-related violence has strongly focused on

religious ideas and socio-political contexts. Although these perspectives are

valuable, they often neglect the way pictures of violence play a role in instigating

and enhancing conflict”.

I have rewritten the beginning of the abstract according to your suggestions. I hope this is a better read now.

  • Images in contemporary society

On page 3, the authors wrote, "On the other hand, however, only a few authors who

have taken seriously this ‘material turn’ within religious studies have also asked

attention for the relationship between materiality, religion, and violence". This

statement above involving, “only a few authors” warrants at least a citation, please.

While a citation is required after the above sentence, Lazarus's (2019) work is also

relevant here. This work is relevant to the current manuscript, not the least because

it demonstrate how images in a contemporary society, influences behavior, and why

people react to them in the ways that they do. For example, the images/figures in

Lazarus’s (2019) paper clearly resonate with Frances King's (2010) words, which the

authors rightly mentioned on page 3 too.

Lazarus’s article is indeed very interesting. Thank you for this! I have made a few references to Lazarus in the text. This article is indeed especially interesting for the question why people respond to pictures and how this relates to offline histories and frames. A powerful case-study.

  • Material objects & aesthetic force

The authors wrote: “Actors’, Bartmanski and Alexander argue, ‘have iconic

consciousness when they experience material objects, not only understanding them

cognitively or evaluating them morally but also feeling their sensual, aesthetic force’

(Bartmanski & Alexander, 2012, 1)”. It could be useful to include recent works such

as Qin & Song’s (2020) study in this discussion here.

Qin and Song is indeed an interesting article. I was not familiar with their research. I included a reference in the text but will certainly use their findings for further research. Also, the reference list they use is very interesting. Thank you!

3

  • Paragraphing….

On page 6, the authors wrote: "[1]....the Portuguese King João III, in 1552, an English

301 Calvinist merchant (A new paragraph is NOT required here, that is, between

these two parts) [2] suddenly leapt from the crowd, punched the elderly..."

Done. Thank you.

  • Proofreading

While the article is generally well written, a careful proofreading would help to

weed out a few grammar errors and long sentences such as the one below:

The determined will we saw in Samudra to destroy two clubs on Bali with hundreds

of human bodies inside of it and the raging ‘zeal’ we saw in the Muntada al-Ansar

video to erase the ‘humiliating images’ of Abu Ghraib by destroying its image with

Nicholas Berg inside of it as an accused ‘American infidel’ contributes heavily to the

iconography of ‘America’.

Thank you. The text has been re-read and long sentences been made more readable.

Reviewer 2 Report

The article is in line with the theme of the special issue.

  1. An in-depth state of the art on the relationship of religion with the media and their impact on material religion in the study of religion-related violence is absolutely necessary. The text lacks this absolutely essential contribution in the context of the meta phenomenon of mediatization/medialization of everything. The article must also discuss the issue from the point of view of the mediatization. An important and recent bibliography on the subject here:

https://www.essachess.com/index.php/jcs/article/view/469/494

https://www.essachess.com/index.php/jcs/issue/view/25

https://journals.sagepub.com/doi/abs/10.1177/0037768616652335

https://www.mdpi.com/2071-1050/12/5/2095

  1. The article lacks methodological details. Even if the appearance is that the text is intended to be a theoretical overview, it is not clear the hypothesis to defend. Details should also be provided on this point.

Author Response

Thank you for reading this article and for your valuable comments. I have added a substantial part on medialization and agree that the first draft missed this specific point of view. I have predominantly focused on the medialization of conflict as this relates to the main argument of this article. I have added relevant literature dealing with the medialization of religious violence. Thank you for the links to medializations of religion. Especially the article from Sbardelotto was interesting.

I have rewritten the first pages and sharpened my hypothesis. Thank you for mentioning that this was not clear yet. I hope with these corrections and improvements the article has become sharper.

Reviewer 3 Report

Minor idiomatic revision suggested, but not imperative.

Author Response

Thank you for reading my article. I have improved the text and eliminated idiomatic mistakes and typos. I also added a section on medialization as that was still a perspective missing in the first draft. 

Round 2

Reviewer 1 Report

I firmly believe the manuscript now warrants publication in Religions: the authors have done an excellent revision.

Once again, I commend the authors for writing this brilliant article.  

Author Response

Thank you again for your inspirational and constructive feedback!

Reviewer 2 Report

Just completing information about the medialization with the French perspective cf Bratosin (2016), “La médialisation du religieux dans la théorie du post néo-protestantisme”, Social Compass, September 63: 405-420, doi:10.1177/0037768616652335

Author Response

Thank you again for your constructive feedback. I have studied the source you mentioned and added it to the article.